# A Combined UWB/IMU Localization Method with Improved CKF

**DOI:** 10.3390/s24103165

**Published:** 2024-05-16

**Authors:** Pengfei Ji, Zhongxing Duan, Weisheng Xu

**Affiliations:** College of Information and Control Engineering, Xi’an University of Architecture and Technology, Xi’an 710311, China; pengfei@xauat.edu.cn (P.J.); xuws@xauat.edu.cn (W.X.)

**Keywords:** ultra-wide band, non-line-of-sight, Levenberg–Marquardt, cubature Kalman filter

## Abstract

Aiming at the problem that ultra-wide band (UWB) cannot be accurately localized in environments with large noise variations and unknown statistical properties, a combinatorial localization method based on improved cubature (CKF) is proposed. First, in order to overcome the problem of inaccurate local approximation or even the inability to converge due to the initial value not being set near the optimal solution in the process of solving the UWB position by the least-squares method, the Levenberg–Marquardt algorithm (L–M) is adopted to optimally solve the UWB position. Secondly, because UWB and IMU information are centrally fused, an adaptive factor is introduced to update the measurement noise covariance matrix in real time to update the observation noise, and the fading factor is added to suppress the filtering divergence to achieve an improvement for the traditional CKF algorithm. Finally, the performance of the proposed combined localization method is verified by field experiments in line-of-sight (LOS) and non-line-of-sight (NLOS) scenarios, respectively. The results show that the proposed method can maintain high localization accuracy in both LOS and NLOS scenarios. Compared with the Extended Kalman filter (EKF), unbiased Kalman filter (UKF), and CKF algorithms, the localization accuracies of the proposed method in NLOS scenarios are improved by 25.2%, 18.3%, and 11.3%, respectively.

## 1. Introduction

With the rapid development of wireless network technology as well as communication technology, there is an increasing demand for positioning services, especially in the fields of indoor pedestrian navigation, robot positioning, and drone positioning. Robot localization plays an indispensable role. However, despite years of research, robot localization is still a challenging problem in practice. In outdoor environments, robots can rely on GPS for high-precision localization [1,2,3]. However, in indoor environments, complex signal propagation environments and building occlusions impose limitations on high-precision robot localization. For this reason, scholars have conducted extensive research on indoor localization technology, which has promoted the rapid development of indoor localization technology.

At present, the commonly used indoor localization techniques include Wi-Fi localization, Bluetooth localization, ultra-wide band (UWB) localization, inertial navigation localization, and visual localization [4,5,6,7]. Compared to several other positioning methods, UWB positioning technology has many advantages. UWB refers to a pulse radio technology with a high bandwidth ratio, which is usually able to provide sub-centimeter positioning accuracy and has a good anti-multipath effect as well as high immunity against interfering signals due to its wide-bandwidth characteristics [8,9,10,11]. Therefore, UWB positioning technology is widely used in indoor positioning. Nevertheless, UWB positioning for indoor complex environments still has shortcomings. The occlusion of indoor buildings interferes with or blocks the transmission of wireless signals, resulting in large non-line-of-sight (NLOS) errors, which makes the positioning performance degradation or even positioning failure. The Inertial Navigation System (INS) adopts kinematic sensors and carries out autonomous localization through its own integrated accelerometers and gyroscopes [12,13,14], and the localization accuracy is not affected by the external environment. However, due to the long-term drift of accelerometers and gyroscopes, the error grows with time. For this reason, some scholars have extensively studied the information fusion scheme of inertial navigation and UWB technology. Peisen et al. [15] provides a UWB/INS loose-combination localization scheme, which employs two parallel Kalman filter (KF)-based models to provide distance estimation by weighted fusion of the filtering results based on the different distance error characteristics of line-of-sight (LOS) and NLOS states. The positioning accuracy of this loose-combination localization system mainly depends on the UWB positioning accuracy, and the algorithm is highly redundant. In the NLOS case, the ability to deal with nonlinear noise through the KF is limited, and the localization results are not good. Zou et al. [16] preprocessed the original UWB measurements using a KF to suppress the UWB distance mutation values and fused the UWB and IMU measurements using an extended Kalman filter (EKF) to adjust the system’s measurement noise covariance matrix in real time, which suppressed the interference of the NLOS effect to a certain extent and improved the localization performance. Narasimhappa et al. [17] designed an improved Sage–Husa adaptive KF, a first-order autoregressive model was used to model the random error of the gyroscope, and the model was used to initialize the transition matrix of the Sage–Husa adaptive KF, and finally, simulations were performed to validate the filter’s performance, which was superior to the KF in removing noise. Li et al. [18] proposed a fusion localization scheme for UWB and IMU based on complementary KF, which utilizes the least-squares method to calculate the localization residuals of UWB observations, determines the robustness factor of the observations, and dynamically sets the observation weights by comparing the magnitude of the robustness factor with the predefined thresholds, which efficiently estimates the observation errors and improves the accuracy of the localization system. Leethter et al. [19] designed an EKF-based fusion localization scheme for UWB and IMU, which simultaneously estimates the systematic error of IMU sensors and corrects the localization error and derived an EKF design for a three-degree-of-freedom planar motion tracking platform, under a specific state of motion; the proposed fusion solution improves 100% over the UWB sensor-based localization scheme in terms of the localization performance. He et al. [20] combined traditional KF with cubature Kalman filtering (CKF) to realize the fusion positioning of UWB and IMU, which makes full use of the characteristic that the measurement equation of the system is linear and uses KF instead of CKF for the computation of the measurement update, which reduces the amount of computation while ensuring the positioning accuracy. Feng et al. [21] performed data fusion of UWB and IMU based on the unscented Kalman filter (UKF) algorithm for single-base station observation and the EKF algorithm for three-base station observation, which reduces the complexity of base station deployment and proposes two approximate motion models to make the localization results smoother, but the proposed fusion algorithms are based on the known statistical properties of the systematic noise and the measurement noise and do not take into account the effect of time-varying noise. Krishnaveni [22] et al. designed an indoor three-dimensional fusion localization system based on UWB and IMU is designed, and the localization data of UWB are solved iteratively by using the time-of-arrival (TOA) method, and the fusion effects of two filtering algorithms, EKF and UKF, are compared, and it is believed that the UKF algorithm can obtain higher localization accuracy in a nonlinear system. However, during the experimental process, the noise distributions of the known statistical characteristics are selected, and the localization performance of the filtering algorithms is not analyzed under the influence of the time-varying noise. In order to solve the non-line-of-sight problem in ultra-wideband ranging of the UWB single-system algorithm, the method of discarding the NLOS information is often adopted, and the disadvantage of this method is that it needs to lay a large number of base stations, which leads to high costs. In the INS single system algorithm, tracking the target for a long period of time leads to the accumulation of drift errors, making the localization fail [23,24,25]. The existing combined UWB and IMU navigation algorithm is based on the processing of UWB outliers when the statistical properties of the noise are known, seldom considers the effect of time-varying noise on the localization system, and does not analyze the localization performance of the system’s filtering algorithm under the influence of time-varying noise.

The environment in which the robot operates is variable, and localization is very challenging due to the presence of indoor buildings and obstacles. First, to address the problem that the initial value is limited during the traditional UWB localization solving process using least-squares, an L–M algorithm is used to optimally solve the UWB position. In addition, considering the adverse effects of measurement noise and NLOS noise on the positioning accuracy, a combined UWB/IMU positioning method with improved CKF is designed. By calculating the residuals between the real measurements and the predicted values, introducing an adaptive factor to update the measurement noise at each moment, and adding a fading factor to suppress the filter divergence, the localization accuracy of the combined indoor robot system is effectively improved.

## 2. UWB Indoor Localization Models and Algorithms

UWB positioning is an indoor high-precision positioning method that achieves accurate position estimation by exploiting the characteristics of ultra-wideband signals [26]. UWB has the advantages of wide bandwidth, narrow pulse, and high time resolution, which are mainly reflected in the TOA-based measurement. Among them, the UWB positioning technique based on TOA of the received signal is widely researched by virtue of its ranging accuracy and high technical feasibility, which basically achieves sub-meter positioning accuracy [27,28]. It calculates the distance by measuring the signal transmission time between the tag and the base station BS1, BS2, and BS3 and then draws a circle centered on the base station to determine the position, as shown in Figure 1.

In practice, however, due to indoor NLOS effects and measurement noise, multiple circles do not intersect at a single point, but rather intersect in a region where the position of the label is in the region. In this case, probabilistic reasoning algorithms and geometric reasoning algorithms are usually used, considering that probabilistic reasoning algorithms require complex mathematical models and dependence on a priori information. In this study, geometric reasoning is used to determine the optimal solution by minimizing the positional differences between multiple circles and labels to obtain the best positional estimate of the labels.

### 2.1. UWB Indoor Localization Model

The channel is an important part of the wireless communication system, and the performance of the wireless communication system depends on the characteristics of the channel. The signal will be affected by the building obstruction during the propagation process, which causes the phenomenon of reflection and refraction of the signal. This results in different times and paths for the signal to reach the receiving end [29]. Therefore, it becomes very important to establish a reliable ultra-wideband model. The commonly used indoor localization models are Poisson model, S-V model, dual-cluster model, and IEEE 802.15.4a channel model. The indoor localization model in this paper adopts the IEEE 802.15.4a channel model, which is characterized by low power consumption, long distance, and high positioning accuracy [30].

For the indoor localization system model, consider that there are *N* pcs UWB base stations and one tag, the location of the tag is (x,y), and the location of the UWB base station is $\left(x_{i},y_{i}\right)$; this study will consider the measurement noise as well as the NLOS effect on indoor localization, modeled as follows:(1)$$R_{i}=d_{i}+\eta_{i}+b_{i}=c\tau_{i},i=1,2,\ldots ,N$$
(2)$$d_{i}=\sqrt{{\left(x-x_{i}\right)}^{2}+{\left(y-y_{i}\right)}^{2}}$$
where $R_{i}$ is the measured distance between the tag and the base station, $d_{i}$ is the true distance between the tag and the base station, $b_{i}$ is the measurement noise, which is modeled here as a zero-mean Gaussian distribution, and $\eta_{i}$ is the positive distance bias introduced by the NLOS effect. The measurement error caused by NLOS in different environments follows gaussian, exponential, and uniform distributions with randomness, non-negativity, and independence [31,32]. Indoors, a mixture of LOS/NLOS generally exists, and in order to reduce the positive distance bias caused by NLOS and improve the accuracy of indoor localization, it has to be taken into account in the modeling process.

### 2.2. UWB Indoor Positioning Algorithms

In the UWB wireless positioning system, according to whether the distance measurement is required for positioning, the positioning methods can be divided into the positioning based on the distance measurement value and the positioning without the distance measurement value. Among them, the positioning methods based on distance measurements are mainly divided into two categories: one is the coordinate position measurement method, which is divided into the trilateral measurement technique, triangulation technique, and great likelihood estimation method; the other is the positioning methods related to position parameters, which is divided into the positioning methods based on the signal arrival TOA, the time difference of arrival (TDOA), based on angle of arrival (AOA) and based on Received Signal Strength Indication (RSSI) [33,34]. TOA can make full use of the characteristics of high time resolution of ultra-wideband signals to detect the signal delay and thus estimate the distance between the node to be located and the reference base station at the receiving end and then calculate the coordinates of the node to be located according to the basic UWB positioning algorithm. In practice, due to the indoor NLOS effect and measurement noise, multiple circles do not intersect at one point, but intersect in a region. Therefore, the problem can be dealt with by the least-squares method to minimize the impact of errors in the ranging process on the localization accuracy. In addition, the Levenberg–Marquardt (L–M) algorithm is an optimization algorithm for nonlinear minimization problems that combines the features of the most rapid descent and the Gaussian Newton method, and it has wide applications in nonlinear regression and parameter estimation problems, the steps of the algorithm are described in Algorithm 1. The method is an improvement on the Gaussian Newton method, which overcomes the problem of inaccurate local approximation or even the failure to converge due to the initial parameters not being set near the optimal solution when solving the traditional least-squares problem.
**Algorithm 1**: L–M algorithm Goal: For a functional relation x=f(p), given f(·) with a noise-laden observation vector x, estimate p. Calculation steps:  Step 1: Take the initial point $p_{0}$, terminate the constant ε, and compute $\varepsilon_{0}=||x-f(p_{0})||$ (which can also be any other number greater than 1). Step 2: Compute the Jacobi matrix $J_{k}$, compute $\bar{N_{k}}=J_{k}^{T}J_{k}+\lambda_{k}I$, and construct the incremental regular equation $\bar{N_{k}}\cdot \delta_{k}=J_{k}^{T}\varepsilon_{k}$. Step 3: Solve the incremental regular equation to obtain $\delta_{k}$.  (1) If $||x-f(p_{k}+\delta_{k})||<\varepsilon_{k}$, then let $p_{k+1}=p_{k}+\delta_{k}$, if $||\delta_{k}||<\varepsilon$, stop the iteration and output the result; otherwise, let $\lambda_{k+1}=\lambda_{k}/\nu$, go to step 2.  (2) If $||x-f(p_{k}+\delta_{k})||\geq \varepsilon_{k}$, then let $\lambda_{k+1}=\gamma \cdot \lambda_{k}$, resolve the regular equation to obtain $\delta_{k}$ and return to step (1).

In this study, we express the ranging error of the circular distance between the function label and the corresponding base station of the first base station as
(3)$$f_{i}(x,y)=\sqrt{{\left(x-x_{i}\right)}^{2}+{\left(y-y_{i}\right)}^{2}}-R_{i}$$

Function $f_{i}(x,y)$ denotes the ranging error of the circular distance between the tag and the corresponding base station of the *i*th base station. The distance vector *F* is defined as $F={\left[f_{1}(x,y),f_{2}(x,y),\cdots ,f_{N}(x,y)\right]}^{T}$. Each element in the *F* vector represents the distance to a circle around the UWB base station, and the Jacobi matrix of *F* representing the differential increment in the position is defined as the gradient of the function:(4)$$J=\nabla_{x,y}[F]=\left[\begin{array}{c} \frac{\partial f_{1}(x,y)}{\partial x}\\ \vdots \\ \frac{\partial f_{N}(x,y)}{\partial x} \end{array}\begin{array}{c} \frac{\partial f_{1}(x,y)}{\partial y}\\ \vdots \\ \frac{\partial f_{N}(x,y)}{\partial y} \end{array}\right]$$

In this study, by defining a cost function ε(x,y) and then using the L–M algorithm to find its minimum value, the method is an improvement made on the basis of Gaussian Newton’s method, which overcomes the problem of inaccurate local approximation or even inability to converge, caused by the fact that the initial parameter is not set near the optimal solution when traditionally solving the least-squares problem.
(5)$$\varepsilon (x,y)=\min\left\{\frac{1}{2}\sum_{i=1}^{N}\beta_{i}{\left[\sqrt{{\left(x-x_{i}\right)}^{2}+{\left(y-y_{i}\right)}^{2}}-R_{i}\right]}^{2}\right\}$$

If one starts at any position $U_{n}={\left[\begin{array}{cc} x_{n} & y_{n} \end{array}\right]}^{T}$, then the position of $U_{n+1}$ is
(6)$$U_{n+1}=U_{n}-{\left(J_{n}^{T}J+\beta diag(J_{n}^{T}J_{n})\right)}^{-1}J_{n}^{T}F_{n},\beta \geq 0$$
where β is the damping factor, and β>0 ensures that the iteration proceeds in the direction of descent. When β is very large, it is close to the most rapid descent method; when β is small, it is close to the Gaussian Newton method.

In order to test the performance of the L–M algorithm, the following simulation experiment is designed: Three base stations in a 30 m × 30 m room with coordinates (10 m, 10 m), (0 m, 15 m), (−5 m, 5 m) are assumed. Where the distance of the UWB mobile tag from these three UWB localization base stations are 15, 10, and 5 m, respectively, using the IEEE802.15.4a channel in which CM4 is implemented as NLOS channel and CM3 is implemented as LOS channel, an exponentially distributed NLOS deviation with a mean value of 2 ns is established for one of the base stations, and the rms delay extension of NLOS satisfies the lognormal distribution with a mean value of 0.021 m and a variance of 0.00178 m. Based on multiple measurements, the thresholds are set to $\delta_{1}=0.3$, $\delta_{2}=0.5$. Finally, the initial position is set to (5 m, 2 m). In summary, this simulation experiment constructs two UWB base stations receiving signals for LOS and one UWB base station receiving signals for NLOS, which is more in line with the actual indoor localization scenario. Finally, the L–M algorithm is used to solve it. The simulation results of the L–M algorithm are shown in Figure 2.

The final solution of (−4.5, 6.3) is obtained through eight iterations. Compared with the actual position, the error is only 0.036, so the position estimation of the UWB tag can be accurately obtained after optimization by L–M algorithm, which further improves the accuracy of the UWB positioning system. Next, the stability of the indoor positioning system will be improved by a fusion filtering algorithm to fuse the positioning information of UWB and IMU.

## 3. UWB/IMU Tight Combination Localization Method

The principle of the combined positioning system scheme in this paper is shown in Figure 3, in which the UWB positioning system consists of four base stations with known coordinates and one tag with unknown coordinates, and the IMU consists of a three-axis accelerometer and a three-axis gyroscope. The acceleration and angular velocity data of the IMU are subjected to an integration operation to obtain the position and velocity information of the IMU. Taking the ranging value of UWB and the yaw angle *φ* of IMU as observation quantities, the improved CKF algorithm is used to centrally fuse the data of UWB and IMU to obtain the velocity and position of the robot.

### 3.1. Combined UWB/IMU Localization Models

Firstly, we define the motion model and observation model of the combined IMU and UWB localization system. The UWB localization model uses the environment to build a coordinate system, and obtains the absolute position information of the UWB tag and the base station through multiple base stations and a single tag, which is called global localization; on the other hand, the IMU localization model builds a coordinate system according to its own motion and uses the data from accelerometers and gyroscopes to obtain the position information of the IMU relative to the robot, which is called local localization. Since the acceleration and angular velocity data of IMU can be obtained by measurement, the state vector of the system is defined as $X={\left[\begin{array}{ccccc} x_{k} & y_{k} & {\dot{x}}_{k} & {\dot{y}}_{k} & \theta_{k} \end{array}\right]}^{T}$. This includes the position, velocity, and heading angle of the robot in the global coordinate system, and the state equation of the system considering the process noise is
(7)$$\left\{\begin{array}{c} x_{k}=x_{k-1}+{\dot{x}}_{k-1}\Delta t+\frac{1}{2}a_{x,k-1}\Delta t^{2}+w_{1k}\\ y_{k}=y_{k-1}+{\dot{y}}_{k-1}\Delta t+\frac{1}{2}a_{y,k-1}\Delta t^{2}+w_{2k}\\ {\dot{x}}_{k}={\dot{x}}_{k-1}+a_{x,k-1}\Delta t+w_{3k}\\ {\dot{y}}_{k}={\dot{y}}_{k-1}+a_{y,k-1}\Delta t+w_{4k}\\ \theta_{k}=\theta_{k-1}+w_{k-1}\Delta t+w_{5k} \end{array}\right.$$
where Δt is the time interval between the robot’s motion from the previous moment to the current moment, θ is the angle between the robot’s human motion direction and the *x*-axis in the global coordinate system, $a_{x,k-1}=a_{bx,k-1}\cos\theta_{k-1}-a_{by,k-1}\sin\theta_{k-1}$ is the robot’s acceleration along the *x*-axis of the global coordinate system, and $a_{bx,k-1}$ and $a_{by,k-1}$ are the robot’s acceleration along the *x*- and *y*-axis of the carrier coordinate system at the *k* – 1 moment, respectively. Similarly, $a_{y,k-1}=a_{bx,k-1}\sin\theta_{k-1}-a_{by,k-1}\cos\theta_{k-1}$ is the acceleration of the robot along the *y*-axis of the global coordinate system at moment *k* – 1. *W*_1*k*_, *w*_2*k*_, …, *w*_5*k*_ for the process noise, the noise vector $W_{k}={\left[\begin{array}{ccccc} w_{1k} & w_{2k} & w_{3k} & w_{4k} & w_{5k} \end{array}\right]}^{T}$ is defined, and Equation (7) is further expressed as
(8)$$X_{k}=FX_{k-1}+BU_{k-1}+W_{k}$$
where
$$F=\left[\begin{array}{ccccc} 1 & 0 & \Delta t & 0 & 0\\ 0 & 1 & 0 & \Delta t & 0\\ 0 & 0 & 1 & 0 & 0\\ 0 & 0 & 0 & 1 & 0\\ 0 & 0 & 0 & 0 & 1 \end{array}\right],\;U_{k-1}=\left[\begin{array}{c} a_{bx,k-1}\\ a_{by,k-1}\\ w_{k-1} \end{array}\right],\;B=\left[\begin{array}{ccc} \frac{1}{2}\cos\theta_{k-1}\Delta t^{2} & -\frac{1}{2}\sin\theta_{k-1}\Delta t^{2} & 0\\ \frac{1}{2}\sin\theta_{k-1}\Delta t^{2} & \frac{1}{2}\cos\theta_{k-1}\Delta t^{2} & 0\\ \cos\theta_{k-1}\Delta t & -\sin\theta_{k-1}\Delta t & 0\\ \sin\theta_{k-1}\Delta t & \cos\theta_{k-1}\Delta t & 0\\ 0 & 0 & \Delta t \end{array}\right]$$
where $X_{k-1}$ is the system state at moment *k* – 1, *F* is the state transfer matrix, *B* is the control input matrix, which is a time-varying matrix based on the heading angle θ, $U_{k-1}$ is the IMU measurement vector, and $W_{k}\sim N(0,Q_{k})$, $Q_{k}$ are the system noise variance matrices.

The nonlinear discrete system is denoted as
(9)$$X_{k}=f(X_{k-1},U_{k-1})+W_{k}$$
(10)$$Z_{k}=h(X_{k})+V_{k}$$
where *X_k_* denotes the state vector at moment *k*, *Z_k_* denotes the observation vector at moment *k*, h(·) is the measure function, measurement noise $V_{k}\sim N(\mu ,R_{k})$, where μ is the mean and $R_{k}$ is the measurement noise variance matrices.

### 3.2. Combinatorial Localization Algorithm Based on Improved CKF

(1) Initialize the state estimates $X_{k}$ and the error covariance matrix $P_{k}$.

(2) Calculate the volume points:(11)$$P_{k}=U\Sigma U^{T}=\left(U\sqrt{\Sigma }\right){\left(U\sqrt{\Sigma }\right)}^{T}=S_{k}S_{k}^{T}$$
(12)$$X_{k}^{j}=S_{k}\xi_{j}+{\hat{X}}_{k},j=1,2,\ldots ,m$$
where *m* is the total number of volume points, according to the ball-radial volume law, m=2n, where *n* is the dimension of the state volume. $\Sigma =diag(\lambda_{1},\lambda_{2},\ldots \lambda_{n})$ is the diagonal matrix consisting of the eigenvalues. $\xi_{j}$ is the volume point, set to
(13)$$\xi_{j}=\left\{\begin{array}{l} \sqrt{n}I_{j},j=1,2,\ldots ,n\\ -\sqrt{n}I_{j}j=n+1,n+2,\ldots ,2n \end{array}\right.$$
where $I_{j}$ denotes the *j*th column of the unit matrix, and the weights of the volume points are set to $w_{j}=1/m$.

(3) By nonlinear state transformation, the volume point is:(14)$$X_{k|k-1}^{j}=f(X_{k-1}^{j},u_{k-1})j=1,2,\ldots ,m$$

(4) The predicted state and error covariance is calculated as follows:(15)$${\hat{X}}_{k|k-1}=\sum_{j=1}^{2n}w_{j}X_{k|k-1}^{j}$$
(16)$$P_{k|k-1}=\sum_{j=1}^{2n}w_{j}X_{k|k-1}^{j}{\left(X_{k|k-1}^{j}\right)}^{T}-{\hat{X}}_{k|k-1}^{j}{\left({\hat{X}}_{k|k-1}^{j}\right)}^{T}+Q_{k}$$

(5) Noise update: Since UWB is subject to uncertain statistical characteristics during ranging and localization, the Sage–Husa noise estimator can estimate and correct the statistical characteristics of the system noise in real time. The Sage–Husa noise estimator can estimate and correct the statistical characteristics of the system noise in real time. When the statistical characteristics of the system noise changes are small, it is necessary to update the measurement noise, so the Sage–Husa noise estimator is used to update the UWB measurement noise as a volume measurement to realize an accurate estimation of the fusion positioning system. The specific steps are as follows:

Denote the measurement prediction for the *j*th sample as
(17)$$X_{k|k-1}^{i}=h(X_{k|k-1}^{i})$$

Calculate the predicted value of the measurement at moment *k*:(18)$${\hat{Z}}_{k|k-1}=\sum_{j=1}^{2n}w_{j}X_{k|k-1}$$

Create prediction error vectors for measurements:(19)$${\tilde{Z}}_{k}=Z_{k}-{\hat{Z}}_{k|k-1}$$

Use the measurement noise estimator considering Sage–Husa filtering based on CKF algorithm characteristics:(20)$${\hat{R}}_{k}=(1-\delta_{k}){\hat{R}}_{k-1}+\delta_{k}({\tilde{Z}}_{k}{\tilde{Z}}_{k}^{T}-H_{k}P_{x,k|k-1}H_{k}^{T})$$
where $\delta_{k}$ is the adaptive factor for Sage–Husa filtering. Since the structural parameter matrix $H_{k}$ does not exist in the CKF algorithm, and since the state vector is high-dimensional in this paper, $\left(1-\delta_{k}\right)$ is rewritten as $\left(I-D_{k}\right)$, which leads to:(21)$${\hat{R}}_{k}={\hat{R}}_{k-1}(I-D_{k})+({\tilde{Z}}_{k}{\tilde{Z}}_{k}^{T}-P_{x,k|k-1})D_{k}$$
where *I* is the unit matrix and $D_{k}$ is the adaptive factor matrix, as specified in the expression $D_{k}=diag(d_{k1},\ldots ,d_{ki},\ldots ,d_{kn}),i=1,2,\ldots ,n$, where the diagonal elements are:(22)$$d_{ki}=\frac{1-b_{i}}{1-b_{i}^{k+1}}$$
where $d_{ki}$ is the adaptive factor, $b_{i}$ is the forgetting factor—which is used to limit the memory length of the filter, where the value range is generally (0.95, 0.99)—and the subscript *n* represents the state dimension. When the statistical characteristics of the noise change quickly, the forgetting factor takes a large value, and when the change is slow, it takes a small value. When the filter converges, the error covariance matrix $P_{x,k}$ will gradually become smaller, and accordingly, $P_{x,k|k-1}$ will also gradually become smaller and eventually converge to 0; then, the measurement noise covariance matrix ${\hat{R}}_{k}$ becomes
(23)$${\hat{R}}_{k}={\hat{R}}_{k-1}(I-D_{k})+{\tilde{Z}}_{k}{\tilde{Z}}_{k}^{T}D_{k}$$

(6) Measurement Updates:

When the UWB localization is in a NLOS scenario, a large variation in the measurements may cause filter divergence. Specifically, if the inequality relation equation holds, the filtering converges and vice versa, then the filtering diverges.
(24)$${\;\tilde{Z}}_{k}^{T}{\tilde{Z}}_{k}\leq \rho \times tr\left[\sum_{j=1}^{2n}w_{j}(X_{k|k-1}^{j}-{\hat{Z}}_{k|k-1}){\left(X_{k|k-1}^{j}-{\hat{Z}}_{k|k-1}\right)}^{T}+{\hat{R}}_{k}\right]$$
where tr(·) is the trace operation of the matrix and ρ≥1 is the adjustable factor [35]. If the filtering diverges, the introduction of the asymptotic cancellation factor $\varphi_{k}$ corrects the one-step prediction covariance matrix $P_{z,k|k-1}$ of the quantitative measurements and induces the filtering to converge. The correction is made by multiplying by the asymptotic factor $\varphi_{k}$ [36]. The corrected expression is
(25)$$P_{z,k|k-1}=\varphi_{k}\sum_{j=1}^{2n}w_{j}(X_{k|k-1}^{j}-{\hat{Z}}_{k|k-1}){\left(X_{k|k-1}^{j}-{\hat{Z}}_{k|k-1}\right)}^{T}+{\hat{R}}_{k}$$
(26)$$\varphi_{k}=\max\left\{1,tr({\tilde{Z}}_{k}{\tilde{Z}}_{k}^{T})/tr(P_{z,k|k-1})\right\}$$

When the UWB localization is in the LOS scenario, the fading factor $\varphi_{k}=1$; only the adaptive matrix $D_{k}$ needs to be adjusted to fit the noise statistics of different sensors. When the UWB localization is in the NLOS scenario and the filtering dispersion, the fading factors $\varphi_{k}>1$ and $P_{z,k|k-1}$ increase, and the filtering gain decreases, which means that the filtering process “trusts” the predicted estimates less, and the filtering results tend to be more in favor of the estimated value of the system state, so as to reduce the impact of the UWB anomalies on the filtering results in the NLOS scenario.

After applying the fading factor correction, the interaction covariance matrix between the state space and the measurement space is expressed as
(27)$$P_{xz,k|k-1}=\varphi_{k}\sum_{j=1}^{2n}w_{j}(X_{k|k-1}^{j}-{\hat{X}}_{k|k-1}){\left(X_{k|k-1}^{j}-{\hat{Z}}_{k|k-1}\right)}^{T}$$

The filter gain is
(28)$$K_{k}=P_{xz,k|k-1}P_{z,k|k-1}^{-1}$$

The estimate of the state of the system at the current *k* moments is
(29)$${\hat{X}}_{k}={\hat{X}}_{k|k-1}+K_{k}(Z_{k}-{\hat{Z}}_{k|k-1})$$

The a posteriori estimate of the updated error covariance matrix is
(30)$$P_{x,k}=\varphi_{k}P_{x,k|k-1}-K_{k}P_{z,k|k-1}K_{k}^{T}$$

In summary, when performing data fusion, the adaptive factor and the fading factor need to be set appropriately to effectively balance the contributions of the state estimates and the actual measurements to the filtered estimates. The above steps (2) to (6) are repeated to obtain the optimal state estimation of the system, and the overall flow is shown in Figure 4.

## 4. Experimental Verification

### 4.1. Experimental Scenario and Equipment

In this section, the proposed combinatorial localization method is further validated through field experiments based on the modeling in the previous section. As shown in Figure 5 and Figure 6, the LOS and NLOS experiments are conducted in a large conference room and a laboratory, respectively. In the LOS scenario, the localization area is kept empty, and no obstacles are placed. In the NLOS scenario, two cardboard boxes were placed in the localization area; in order to make the experiment more closely resemble a real indoor NLOS scenario, we placed five 10-cm-thick wooden boards in one of the cardboard boxes. In the other cardboard box, multiple bricks of concrete material were placed, totaling about 50 cm in thickness. During the experiments, in the LOS environment, the robot moves along a preset rectangular trajectory and collects UWB and IMU localization information to verify the localization accuracy of several filtering algorithms. In the NLOS environment, static and dynamic experiments are performed separately. In the static experiments, the robot is placed in the center of the localization area and remains stationary, and obstacles randomly block the propagation of UWB wireless signals. In the dynamic experiments, the robot does not preset the walking trajectory and moves around the obstacles in the localization area, taking the localization data collected by the high-precision LiDAR as the reference trajectory. Finally, several filtering algorithms are used to obtain the combined UWB and IMU localization trajectories in static and dynamic environments, respectively, to verify the performance of the algorithms.

In this paper, the proposed method is tested using an ultra-wideband model based on IEEE 802.15.4a [29]. The UWB device is a DW1000 chip from DecaWave in Dublin, Ireland, which is used to collect the position information of the robot, and the built-in IMU sensor with model number ICM-42605 (TDK-InvenSense co., San Jose, CA, USA) is used to collect the acceleration and angular speed information of the robot. The UWB and IMU devices selected for the experiment are shown in Figure 7a, and the mobile robot is shown in Figure 7b. The sampling frequencies of UWB and IMU are 50 Hz and 200 Hz, respectively. Before the experiment, the ranging data of UWB are pre-filtered once, and the zero bias of the IMU are recorded to compensate for the sampled values of subsequent IMUs.

### 4.2. LOS Scenario Experiments

The experimental environment is a large conference room of 20 m × 40 m, with no obstacles blocking it, which can be regarded as a LOS environment. Before the experiment, we recorded the zero-bias value of the IMU, in order to compensate for the subsequent sampling data and reduce the effect of accelerometer drift on the IMU positioning error. For long-time positioning, we calibrated the accelerometer and Gyroscope periodically, which can eliminate part of the error accumulation and improve the measurement accuracy. The robot moves at a speed of 0.06 m/s, and four UWB base stations simultaneously range the robot’s trajectory points. The real trajectory of the robot is a rectangular area of 13 m × 27 m, and the robot moves according to the preset trajectory during the experiment. The initial position coordinates of the robot are (2.5, 6), and the motion planning of the robot is realized by the spatial programming of the robot operating system under the Ubuntu system, which outputs the rectangular trajectory, and the four vertices of the motion trajectory area have the coordinates of (2.5, 6), (15.5, 6), (2.5, 33), (15.5, 33). Finally, the position information of the UWB and IMU is measured separately, and their localization trajectories are shown in Figure 8.

Figure 8 shows the localization trajectories of IMU and UWB in an indoor LOS environment. In the LOS environment, the UWB localization trajectory is close to the actual trajectory, and the localization accuracy is very high; the IMU has a very high accuracy at the initial stage of localization, and there is no significant difference compared with the UWB. Over time, the positioning accuracy of IMU decreases. We visualize the relevant parameters of the accelerometer and gyroscope in the IMU. From Figure 9, it can be seen that there is a small drift in the acceleration and angular velocity information obtained in the accelerometer and gyroscope, and with the passage of time, the cumulative error will be generated after the integration operation, and the positioning accuracy will be degraded. Therefore, the filtering algorithm can be used to combine the UWB and IMU for positioning, correcting the position of the IMU by using the feature of high positioning accuracy of the UWB and assisting the UWB positioning by using the feature of the IMU that is not affected by the environment.

In order to verify the localization performance of the proposed improved CKF algorithm in indoor LOS environments, we conducted combined localization experiments, and the experimental site is shown in Figure 5. The mobile robot carries UWB tags and IMUs along the preset trajectory and fuses the position information of UWBs and IMUs using EKF, UKF, CKF, and improved CKF algorithms, respectively. The combined localization trajectory is shown in Figure 10.

As can be seen from Figure 10, EKF has the largest localization error, followed by UKF. Compared to these two algorithms, CKF has a lower localization error; this is due to the fact that CKF uses cubic coordinate transformations to deal with the nonlinear system, by averaging in a higher-dimensional space, avoiding the process of linearization of the nonlinear function. CKF is similar to UKF in that it deals with the nonlinear system by mapping the sampling points to a higher dimensional space, but CKF chooses cubic roots as sampling points, which are more characteristic compared to the Sigma points of UKF; therefore, CKF is more rigorous and stable than UKF. Consistently, in the LOS scenario, the velocity error and position error of the improved CKF algorithm in the *x*- and *y*-axis directions are always within a small range, which has certain advantages compared with the EKF and UKF algorithms, and its ability of predicting the noise and suppressing the error is stronger compared with the CKF algorithm. In addition, the improved CKF method proposed in this paper is closest to the real trajectory, smoother than the localization trajectory of the CKF algorithm, and has the highest localization accuracy.

### 4.3. NLOS Scenario Experiments

UWB can satisfy the high-precision localization of indoor robots in LOS environments, but in complex indoor environments, UWB will be affected by factors such as building occlusion, which produces serious NLOS effects and leads to increased localization errors or even failure. In order to verify the localization effect of the proposed improved CKF combinatorial localization algorithm in complex indoor environments, robot localization and navigation experiments are conducted. The experimental setup is consistent with the NLOS experimental scenario described in Section 4.1. In the NLOS scenario, static and dynamic experiments were conducted separately. Here, a “static” experiment refers to one where the robot does not move during the whole data acquisition process, and several filtering algorithms are used to fuse the position information from the UWB and IMU under the condition of maintaining a stationary state. The scatter plot in Figure 11a is a 2D visual representation of the performance comparison of the four localization algorithms in a static scenario. To be more intuitive, the reference position of the static tag is moved to the coordinate origin (0, 0). In this static scene, it can be observed that the localization errors of several algorithms are all within 0.1 m, among which the localization results of EKF algorithm and UKF algorithm have a larger deviation from the real value, while the localization results of CKF algorithm and the improved CKF algorithm are closer to the origin of the coordinates, and the localization errors are smaller.

In order to better compare the localization results of the algorithms, Figure 11b shows the root-mean-square error (RMSE) of the four algorithms for 700 localization samples in a complex indoor environment, which is about 0.032 m for UKF, 0.05 m for EKF, 0.023 m for CKF, and 0.009 m for the improved CKF algorithm. The experimental results show that the proposed improved CKF algorithm has higher localization accuracy in static scenarios and can better meet the requirements of indoor robot localization. Table 1 gives the maximum and average errors of several algorithms in the *x*-direction and *y*-direction.

To verify the localization performance of the proposed algorithm in dynamic environments, we conducted experiments using a mobile robot. The robot carries UWB tags and IMU to move in the localization area and obtains the combined localization trajectories using EKF, UKF, CKF, and modified CKF algorithms. The localization trajectories are shown in Figure 12a. From the figure, it can be seen that the localization errors of EKF and UKF are larger in dynamic environments. When the NLOS effect is severe, the localization accuracy of the EKF, UKF, and CKF algorithms decreases even more. The improved CKF algorithm filters the whole trajectory more closely to the real trajectory and can still maintain good localization accuracy even when the localization results of other algorithms change drastically. This indicates that when the residual difference between the predicted value and the actual value of the observed quantity is too large, the improved CKF algorithm reduces the weight of the measured value, and the influence of the NLOS error on the estimation of the system is reduced by reasonably controlling the gain. The improved CKF algorithm’s trajectory is smoother than that of the CKF algorithm, which overall better balances the contributions of the state and quantity measurements to the filtered estimation.

In order to better compare the localization results of several algorithms, the root-mean-square error of each algorithm is given in Figure 12b. The RMSE error of the EKF algorithm is about 0.38 m, the RMSE error of the UKF algorithm is about 0.35 m, the RMSE error of the CKF algorithm is about 0.25 m, and the RMSE error of the improved CKF algorithm is about 0.13 m. Among these algorithms, the root-mean-square error of the CKF tight combination algorithm is less than several other algorithms. Table 2 gives the maximum localization error and the average error of each algorithm in the *x*-direction and *y*-direction. It can be seen that in complex indoor environments, the localization errors of the CKF tight combination algorithm are significantly smaller than those of the other algorithms in both the *x*- and *y*-directions. In the indoor NLOS environment, the localization accuracy is improved by 25.2%, 18.3%, and 11.3% compared to the EKF, UKF, and CKF algorithms, respectively. It can be seen that the improved CKF algorithm can effectively control the ranging error and ranging outliers in the NLOS scenario of UWB.

## 5. Conclusions

In this paper, taking an indoor mobile robot as the research object, UWB and IMU information were effectively fused, the L–M algorithm was used to optimally solve the UWB position, and an improved CKF combinatorial localization algorithm based on real-time updating of measurement noise was designed. The Sage–Husa filter is used to estimate the statistical characteristics of the updated noise, the noise statistical characteristics of different sensors are adaptively updated, and the fading factor is introduced to prevent filter dispersion and improve filtering speed, which realizes the precise positioning of the mobile robot. Finally, the combined localization performance of this paper’s algorithm is analyzed and evaluated through experiments. The experimental results show that the improved CKF algorithm stabilizes the localization error within 0.1 m and 0.3 m in LOS and NLOS scenarios, respectively. In the NLOS scenario, the localization accuracies of the improved CKF algorithm are 25.2%, 18.3%, and 11.3% higher than those of the EKF, UKF, and CKF algorithms, respectively. The improved CKF algorithm shows a more obvious advantage both in the LOS scenario and in the NLOS scenario.

Our future work will focus on the following aspects:(1)Specific engineering practice often exists in a variety of noise, when the system process noise is complex and exhibits time-varying effects; at this time, it is necessary to comprehensively consider the system process noise and time-varying effects on the fusion localization system, in order to obtain a more accurate position estimation.(2)For UWB and IMU, the localization data were collected under relatively stable motion processes of the mobile robot, and for more complex motion modes, such as rapid acceleration and emergency stop, a better motion state model is needed to fit the tested motion modes. Therefore, it is worthwhile to further investigate how to design a state model that is applicable to motion in multiple modes.

## Figures and Tables

**Figure 1 sensors-24-03165-f001:**
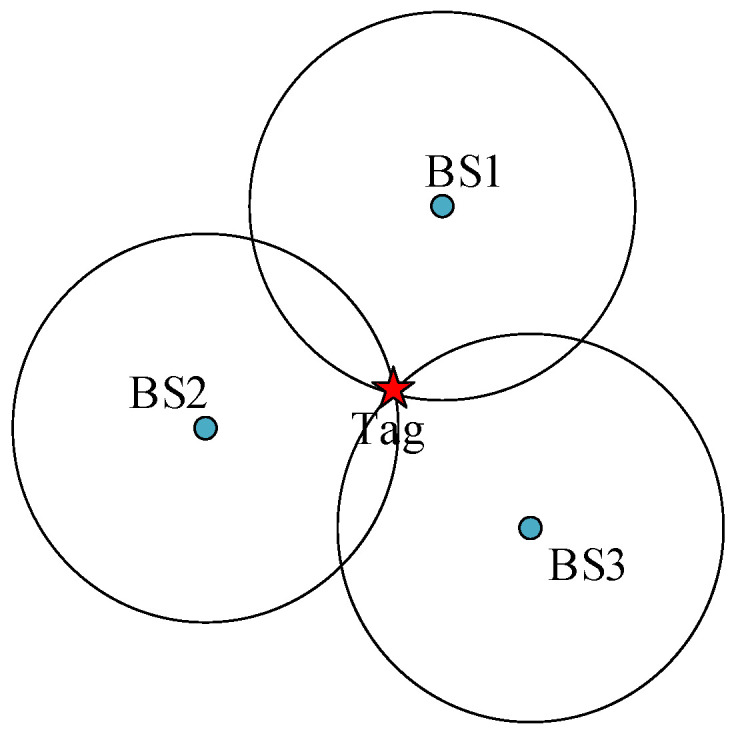
Trilateral positioning method.

**Figure 2 sensors-24-03165-f002:**
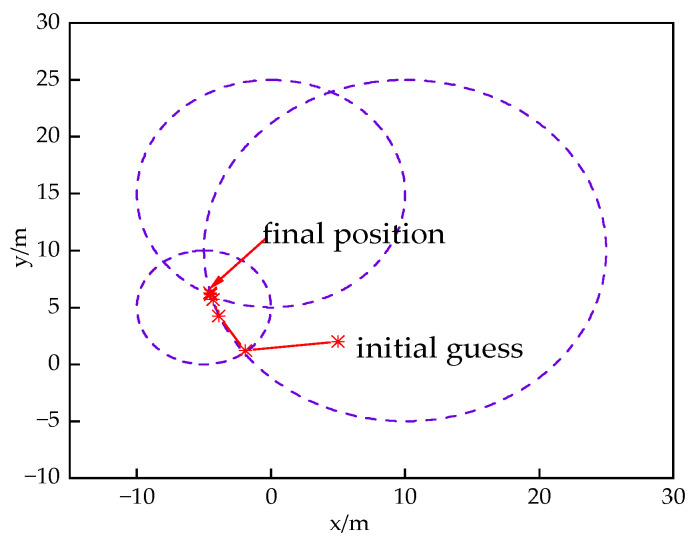
L–M algorithm simulation experiments.

**Figure 3 sensors-24-03165-f003:**
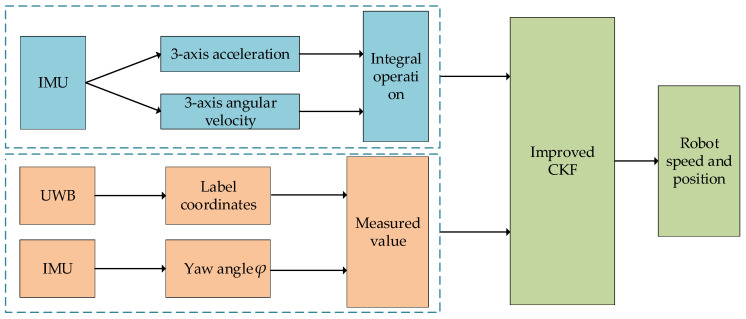
Block diagram of combined UWB and IMU localization.

**Figure 4 sensors-24-03165-f004:**
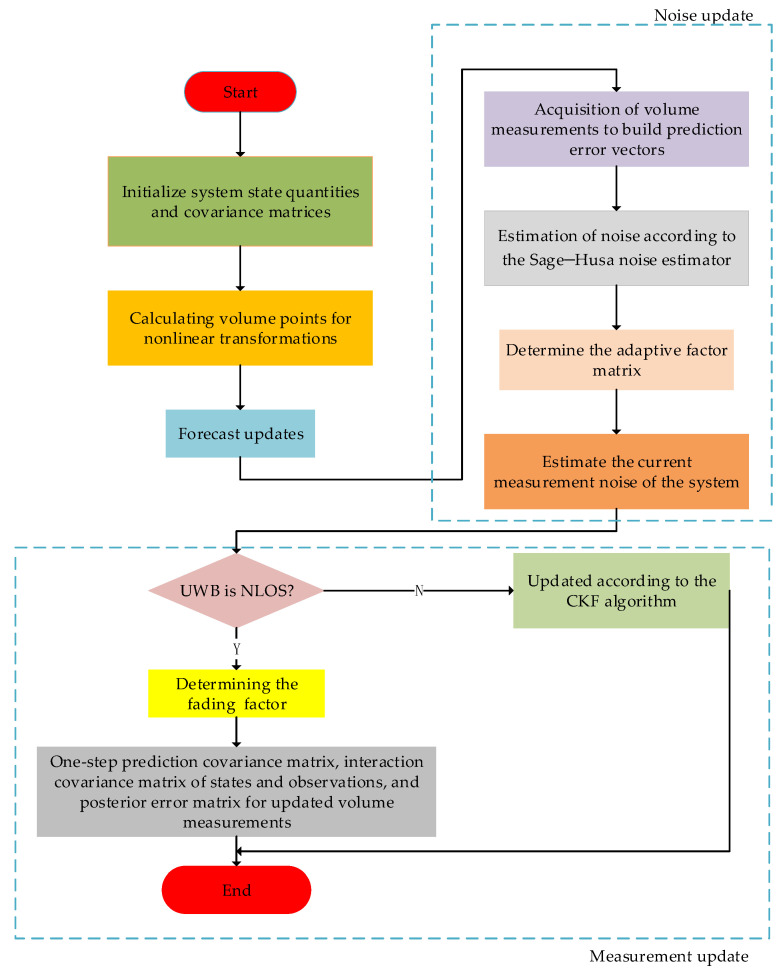
Flowchart of combinatorial localization based on improved CKF algorithm.

**Figure 5 sensors-24-03165-f005:**
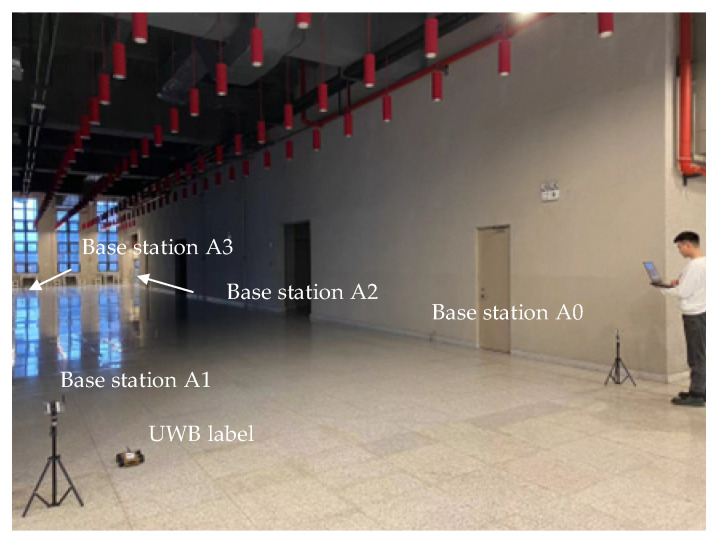
LOS experimental scenarios.

**Figure 6 sensors-24-03165-f006:**
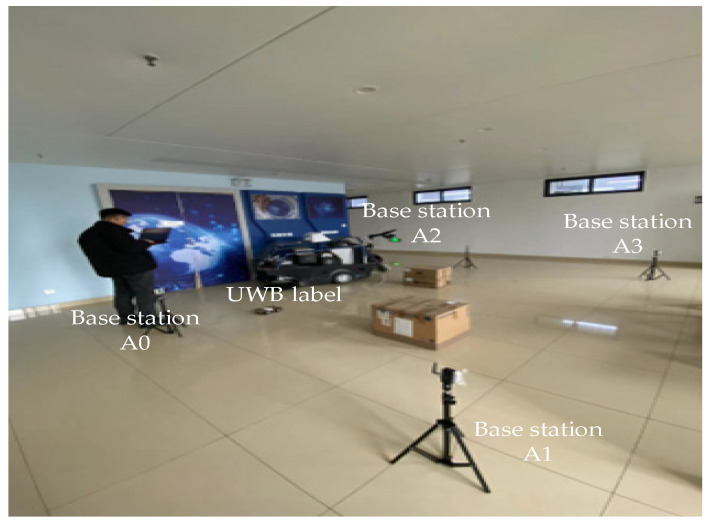
NLOS experimental scenarios.

**Figure 7 sensors-24-03165-f007:**
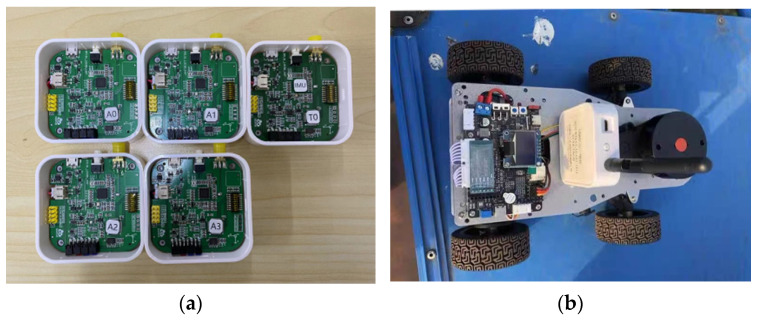
Experimental equipment. (**a**) UWB base stations and tags. (**b**) Mobile robot.

**Figure 8 sensors-24-03165-f008:**
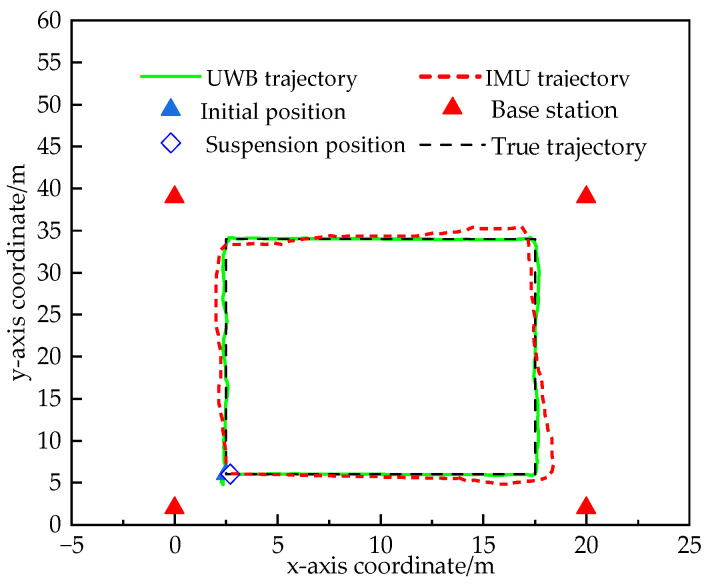
Localization trajectories for UWB and IMU in LOS environments.

**Figure 9 sensors-24-03165-f009:**
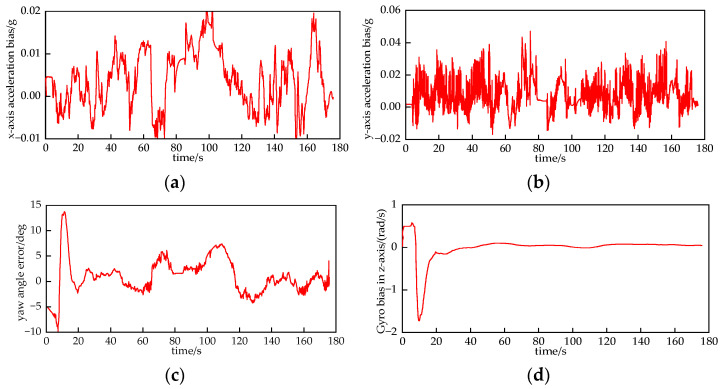
Parameters related to accelerometers and gyroscopes in IMUs. (**a**) Acceleration bias in the *x*-axis. (**b**) Acceleration bias in the *y*-axis. (**c**) Yaw angle error. (**d**) Gyroscope bias in *z*-axis.

**Figure 10 sensors-24-03165-f010:**
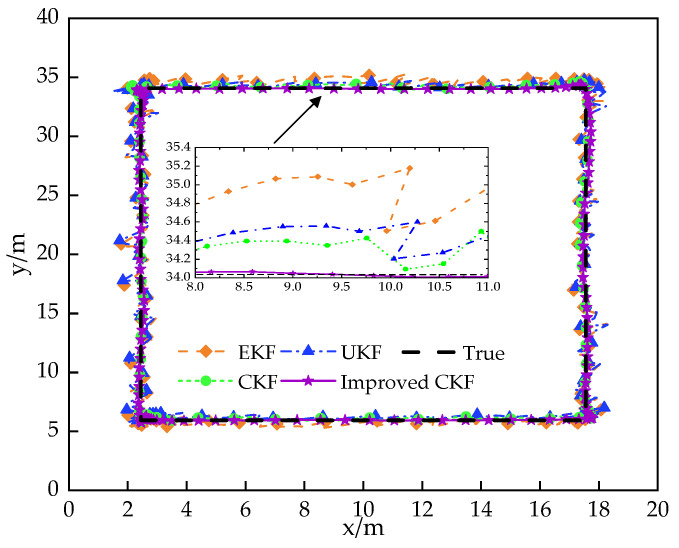
Combined localization trajectories of several algorithms in the LOS environment.

**Figure 11 sensors-24-03165-f011:**
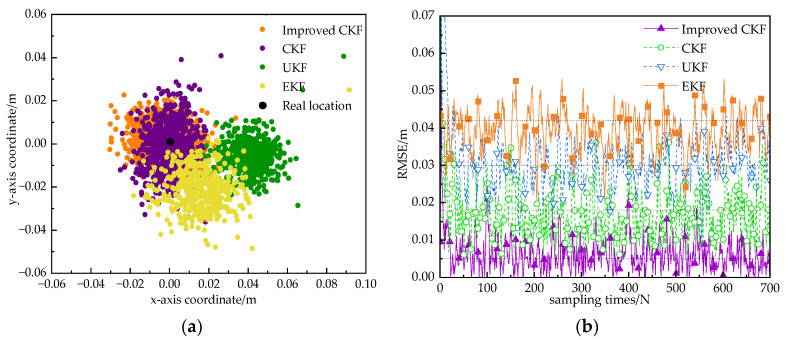
Static 2D localization trajectories and RMSE for several algorithms in the NLOS environment. (**a**) Positioning track. (**b**) RMSE.

**Figure 12 sensors-24-03165-f012:**
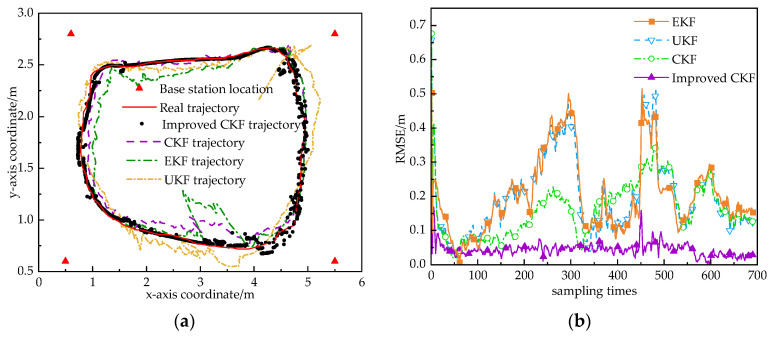
Dynamic 2D localization trajectories and RMSE for several algorithms in NLOS environment. (**a**) Positioning track. (**b**) RMSE.

**Table 1 sensors-24-03165-t001:** Comparison of localization errors in static scenes.

Algorithm	*x*-Direction		*y*-Direction	
	Maximum Error	Average Error	Maximum Error	Average Error
Improved CKF	0.049	0.005	0.061	0.004
CKF	0.057	0.01	0.057	0.02
UKF	0.067	0.03	0.056	0.024
EKF	0.106	0.042	0.056	0.036

**Table 2 sensors-24-03165-t002:** Comparison of localization errors in dynamic scenes.

Algorithm	*x*-Direction		*y*-Direction	
	Maximum Error	Average Error	Maximum Error	Average Error
Improved CKF	0.37	0.13	0.62	0.14
CKF	0.69	0.24	0.61	0.22
UKF	0.76	0.34	0.85	0.34
EKF	0.89	0.35	0.74	0.33

## Data Availability

Data are contained within the article.

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
