# Peer review of "A Combined UWB/IMU Localization Method with Improved CKF"

_sensors, 2024, doi:10.3390/s24103165_

Round 1

Reviewer 1 Report

Comments and Suggestions for Authors

1. The references are totally irrelevant.

Some examples:

3.  Wang, K.; Pang, L.; Li, X. Identification of Stopping Points in GPS Trajectories by Two-Step Clustering Based on DPCC with 503

Temporal and Entropy Constraints. Sensors 2023, 23, 3749. 

4. Joshi, S.; Kannaujiya, S.; Joshi, U. Analysis of GNSS Data for Earthquake Precursor Studies Using IONOLAB-TEC in the Hima- 505

layan Region. Quaternary 2023, 6, 27. 

5.  Spravil, J.; Hemminghaus, C.; von Rechenberg, M.; Padilla, E.; Bauer, J. Detecting Maritime GPS Spoofing Attacks Based on 507

NMEA Sentence Integrity Monitoring. J. Mar. Sci. Eng. 2023, 11, 928. 

These papers are about GNSS,  not indoor technologies

6. Yang, Z.D.; Liu, W.L.; Zhang, J.F.; Zhang, X.; Zhao, C.; Liu, Y.C. Research on auto drive system of Beidou Navigation Agricul- 509

tural Machinery. Sci. Technol. Innov. 2021, 20, 47–49.

7. Bian, S.F.; Liu, Y.; Ji, B.; Zhou, W. Statistical Characteristics Analysis of Altitude Angle Correlation Stochastic Model of Beidou- 511

3 Satellite Observation Information. Geomat. Inf. Sci. Wuhan Univ. 2022, 47, 1615–1624.

Can't be found, probably local journals, not about UWB

11. Park, J.S.; Lee, B.; Park, S.; Kim, C.H. Estimation of Stride Length, Foot Clearance, and Foot Progression Angle Using UWB 519

Sensors. Appl. Sci. 2023, 13, 4801.

This paper is mainly about UWB and its comparison with IMU, not about IMU. 

13. Kolangiammal, S.; Balaji, L.; Mahdal, M. A Compact Planar Monopole UWB MIMO Antenna for Short-Range Indoor Applica- 523

tions. Sensors 2023, 23, 4225.

This paper is about antenna, not about a UWB/INS loose-combination localization scheme

16. Hiroki, K.; Masaki, O.; Takeshi, N.; Takeshi, Y.; Takanori, F. Localization Method Using Camera and LiDAR and its Application 529

to Autonomous Mowing in Orchards. J. Robot. Mechatron. 2022, 34, 877–886.

17. Salmane, P.H.; Rivera Velázquez, J.M.; Khoudour, L.; Mai, N.A.M.; Duthon, P.; Crouzil, A.; Pierre, G.S.; Velastin, S.A. 3D Object 531

Detection for Self-Driving Cars Using Video and LiDAR: An Ablation Study. Sensors 2023, 23, 3223. 

18. Hu,Z.; Mao, J.D.; Zhou, C.Y.; Gong, X. A method of determining multi-wavelength lidar ratios combining aerodynamic particle 533

sizer spectrometer and sun-photometer. J. Quant. Spectrosc. Radiat. Transf. 2018, 217, 224–228.

These papers are about lidars, not about INS or drift errors

23. LI, C; FU, Y; YU, F. Vehicle position correction: A vehicular blockchain networks-based GPS error sharing framework[J]. IEEE 543

Transactions on Intelligent Transportation Systems, 2022, 22(2): 898-912

This paper is about GPS, not about IEEE 802.15.4a channel model

27. Radha, S.M.; Jung, M.; Park, P.; Yoon, I.-J. Design of an Electrically Small, Planar Quasi-Isotropic Antenna for Enhancement of 552

Wireless Link Reliability under NLOS Channels. Appl. Sci. 2020, 10, 6204.

This paper is about antenna, not about classification of positioning technologies. Many surveys exist that can be used as reference in this context.

Some paper in Chinese and can't be checked for relevance.

1. Dong, S.; Yuan, C.H.; Gu, C.; Yang, F.; Fu, H.; Wang, C.; Jin, C.; Yu, J. Research on intelligent agricultural machinery control 499

platform based on multi-discipline technology integration. Trans. Chin. Soc. Agric. Eng. (Trans. CSAE) 2017, 33, 1–11.

19. Zhao,Z.Y.; Feng, Z.K.; Tian, Y.; Liu, J. Design and test of photographic dendrometer based on Ultra-Wide Band (UWB) posi- 535

tioning. Trans. Chin. Soc. Agric. Eng. (Trans. CSAE) 2020, 36, 167–173.

21. Sheng, K.P.; Wang, J.; Li, C.H.; Li, X.G. Research on ultra-wideband positioning model for non-line-of-sight error correction. J. 539

Sci. Surve. Map. 2021, 46, 40–47.

12. Wen, K. Research on the Key Techniques of Ultra-Wide band Based Indoor Position and Orientation Estimation. Master’s Thesis,

Wuhan University, Wuhan, China, 2020

Not available

8. Ma, M.B.; Tang, J.S.; Tian, Z.; Chen, Z.P. Motion compensation of multiple-receiver synthetic aperture sonar based on high

precision inertial navigation system. Huazhong Univ. Sci. Tech. 2020, 48, 73–78. 

Not available

------------------------------------------

2. You write that the robot moved along the intended path for one week. Why in Fig. 8 UWB has only one trajectory? Has the robot made one circle in a week? What is the speed of the robot?

Or is this the result of averaging trajectories? Then how was it obtained?

Why in Fig. 8 IMU has such a strange trajectory if it is installed on a robot?

How do you deal with accelerometer drift over such a long time?

------------------------------------------

3. Image 9 has poor quality.

------------------------------------------

4. Cardboard boxes as a obstacle to radio waves look rather strange. Since the attenuation and reflectivity in the operating range of such systems are small. If special material was used, you need to write about it.

------------------------------------------

5. In Fig. 11a the real trajectory is not visible. Therefore, it is not clear: did the robot stand at one point all the time or did it move? What is shown in the figure: errors at different points or a scatter of values at one point?

In Fig. 11b “700 localization samples” are errors at 700 different points or 700 measurements at one point? If the first, then how many measurements were there at one point? If the latter, how were the RMSE values obtained? What was averaged?

------------------------------------------

6. It is necessary to add references to the algorithms used in the work (EKF, UKF, CKF, improved CKF)

Reviewer 2 Report

Comments and Suggestions for Authors

The authors of the presented work proposed an optimal solution for determining the position of an indoor mobile robot based on information from UWB and IMU. They used the L-M algorithm and an improved CKF combinatorial localization algorithm based on real-time measurement noise update. They used the Sage-Husa filter to estimate the statistical characteristics of the noise. They also experimented to verify the proposed algorithms in both LOS and NLOS scenarios. I consider the topic of the submitted contribution to be current. The mentioned algorithms are correct.  I recommend clarifying the basis for setting the adaptive and fading factors when performing data fusion.  I have no fundamental comments regarding the results of the experiment. The conclusion of the paper should be supplemented with the results of the experiment and the advantages and disadvantages of the mentioned algorithms.The formal administration of the contribution is at the required level. I recommend publishing the post after a slight edit.

Comments on the Quality of English Language

No.

Reviewer 3 Report

Comments and Suggestions for Authors

The manuscript “A combined UWB/IMU localization method with improved CKF” addresses a topic of interest to a broad audience and fits the scope of the journal.

The article is well-organized, particularly in the way it logically progresses from introducing the problem, discussing existing methods, and proposing your improved CKF algorithm. These comments aim to assist in refining the article for publication, ensuring clarity, depth, and a robust presentation of the research findings.

 Here are some review comments and suggestions:

1.     Line 121 What is “N UWB ”? It should be N pcs UBW stations.

2.     Figures and tables are generally well-crafted. Ensure that each figure and table is referenced in the text and discussed adequately to guide the reader through the data. Figure legends should be descriptive enough to be understood independently of the main text.

3. The description of Figure 3 on lines 203-209 is not consistent and hard to understand. Please rewrite the description or redraw Figure 3.

4. In Figure 4, red circles are placed outside the blue frame to describe the frame noise update or measurements update.

5. Line 383 and Line 436: Why did it need one week to derive the localization information of UWB and IMU?

6.     Figure 11 (a) The real trajectory can not be seen in this figure.

Consider elaborating on the experimental setup, particularly the choice of environments for the LOS and NLOS scenarios. Providing more details about the conditions under which tests were conducted would enhance the reproducibility of the results.

Round 2

Reviewer 1 Report

Comments and Suggestions for Authors

The reviewer's comments have been taken into account in the new version of the article.

Text proofreading required. Can be performed at the text editing stage.